# Adsorption Kinetics and Mechanism of Pb(II) and Cd(II) Adsorption in Water through Oxidized Multiwalled Carbon Nanotubes

Xin Li, Yating Cui, Wanting Du, Weiheng Cui, Lijuan Huo and Hongfang Liu *

College of Environment and Resources, Taiyuan University of Science and Technology, Taiyuan 030024, China; 18834173262@163.com (X.L.); 18834725298@163.com (Y.C.); 15137187008@163.com (W.D.); 17799997899@163.com (W.C.); 2006031@tyust.edu.cn (L.H.)
* Correspondence: liu_hf@163.com

**Abstract:** Toxic heavy metals are ubiquitous in the aquatic environment and show a significant danger to human health. Carbon nanotubes have been extensively used in treating the contamination of groundwater due to their porous multi-layer nature. Batch tests revealed that oxidized multiwalled carbon nanotubes (O-MWCNT$_S$) offer better removal of Pb(II). The removal rate of Pb(II) was 90.15% at pH 6 within 24 h, which was ~58% more than that of Cd(II). The removal rate decreased to 55.59% for Pb(II) and to 16.68% for Cd(II) when the initial concentration of Pb(II)/Cd(II) ranged from 5 to 15 mg·g$^{-1}$. The removal rate in the competitive tests was about 60.46% for Pb(II) and 9.70% for Cd(II). The Langmuir model offered better description of the adsorptive data for both ions. And the Q$_m$ of Pb(II) was 5.73 mg·g$^{-1}$, which was 2.39 mg·g$^{-1}$ more than that of Cd(II) in a single-icon system, while Q$_m$ was 7.11 mg·g$^{-1}$ with Pb(II) and 0.78 mg·g$^{-1}$ with Cd(II) in competitive water. And thermodynamic tests further indicated that the activating energy of Pb(II) and Cd(II) was 83.68 and 172.88 kJ·mol$^{-1}$, respectively. Lead and cadmium adsorbed on the surface of O-MWCNT$_S$ are antagonistic in the competitive system. Based on XPS analyses, it was concluded that the absorbed lead/cadmium species on O-MWCNT$_S$ were (-COO)$_2$Pb, (-COO)Pb(-O)/(-COO)$_2$Cd, and (-COO)Cd(-O). Additionally, they offered theoretical evidence supporting the practicality of using nanocomposite membranes as a means to remove cadmium and lead.

**Keywords:** multi-walled carbon nanotubes; adsorption; oxidation; Cd(II); Pb(II)

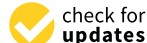



## 1. Introduction

Water is a vital natural resource essential for sustaining human life. Regrettably, the escalation of population development and industrialization has led to the introduction of various pollutants, including heavy metals, dyes, and antibiotics, into pollution-free water environments [1]. Wastewater consists of an assortment of contaminants, some of which are possibly harmful to aquatic life, human beings, and animals. The presence of heavy metals, including Cd(II), Cu(II), Pb(II), and Ni(II), is continually acknowledged around the globe as an increasing menace to the health of humans due to their perilous nature, mobility, non-biodegradability, and propensity for building up in organs such as liver and kidneys [2,3]. Hazardous heavy metals, including lead and cadmium, are commonly encountered in aquatic environments and present a substantial risk to the health of aquatic organisms. It has been reported that Cd(II) poses significant injury to the human bone, the kidneys, and lung capacity, and it is also cancerous [4]. It is known to all that Pb(II) has been linked to disorders of the brain, malignancy, anorexia, plumbism, anemia, renal damage, and dementia [5]. In view of these difficulties, there is a growing need to treat water that is lead-and-cadmium-contaminated throughout the world. In order to do this, the World Health Organization (WHO) proposed the permissible levels of 0.01 and 0.1 mg·L$^{-1}$ for Cd(II) and Pb(II) ions in potable water [6]. Most importantly, it is critical to obtain more

economical and effective treatment methods to mitigate the severe heavy metal hazard to human health and safety [7].

In recent studies, scientists have investigated lots of methods for eliminating Pb(II) and Cd(II), for instance, ion exchanges, chemical precipitates, reversed seepage, membranes processes, microbiological biotechnology, coagulation, flocculation, and filtering and adsorption technologies [8,9]. Nevertheless, the technologies have several disadvantages, namely, a low efficiency, a high cost, a hazardous co-product induction, operational delays, pollutant-specific inefficiencies, and a treatment process complexity [10,11]. Among these techniques, adsorption is considered to be one of the most effectively and widely used techniques. This is primarily owing to its simplicity of operation management, potential for regeneration, cost efficiency, inertness to materials, absence of muck formation, and availability of a wide range of adsorbents [12]. In recent years, there has been a rising interest in the research on nanotechnology because of its unique physical properties, which may be attributed to their small dimensions, shapes, sizes, large surfaces areas, crystallization, and contents. In addition, nanocomposite membranes are considered to be the future of membranes and have demonstrated their remarkable capacity to adsorb heavy metal ions. This ability is greatly influenced by the careful selection of inorganic materials based on their physical properties [13,14].

From a previous review of studies on the removal of toxic metals using carbon nanomaterials, the maximum adsorption capacity for the removal of toxic metals using carbon nanomaterials is in the following order: carbon nanotubes > graphene > activated carbon > carbon dots [15]. In nanomaterials, multi-walled carbon nanotubes ($MWCNT_S$), innovative carbon nanomaterials, have been widely used in the treatment of water contaminated by heavy metals [16]. $MWCNT_S$ are known for their exceptional chemical stability, uniform surface distribution, and porous multilayer structure. However, $MWCNT_S$ tend to agglomerate rapidly, which leads to an obvious decrease in their adsorption efficacy. Previous studies have demonstrated that the solubility and adsorption capacity of MWCNTs can be improved via chemical oxidation [17,18]. Li et al. reported that after $MWCNT_S$ were oxidized with $KMnO_4$, their adsorption capacity for Cd(II) increased by 90% [19]. Furthermore, Lin et al. demonstrated the proportions of hydrophilic functional groups present on the surface of $MWCNT_S$ which were modified with oxidants [20–22]. And these materials modified with the method of oxidation are named $O\text{-}MWCNT_S$ [17,20,21].

The goal of this present work was to preliminary test the effectiveness of $O\text{-}MWCNT_S$ for the removal of Pb(II) /Cd(II) in water. The specific objectives were as follows: (1) to examine the characteristics of $O\text{-}MWCNT_S$ prepared with hydrogen peroxide; (2) to determine the effects of various parameters such as pH, temperature, and initial concentration on the removal efficacy of metal ions Pb(II)/Cd(II); and to (3) identify the absorptive products lead/cadmium on the $O\text{-}MWCNT_S$ and elucidate the underlying mechanisms of adsorption.

## 2. Materials and Methods

### 2.1. Materials and Instruments

2.1.1. Materials

Each chemical reagent utilized in this investigation was analytically pure. NaOH, HCl, $H_2O_2$, $PbNO_3$, and $CdCl_2$ were purchased from Sinopharm Chemical Reagent Co. (Shanghai, China).

2.1.2. Instruments

The apparatus used for the experiment is shown in Table 1.

**Table 1.** Experimental instruments.

| Instrument Name | Model Number | Factory Owners | Type of Analysis |
| --- | --- | --- | --- |
| Electronic Analytical Balance | CPA225D | Sartorius, Göttingen, Germany | Weigh |
| Magnetic heating stirrer | MYP84-1 | Changzhou Ronghua Instrument Manufacturing Co., Changzhou, China | Stirring |
| pH meter | PHS-3C | Mettler Toledo Instruments Ltd., Shanghai, China | pH |
| Inductively Coupled Plasma Emission Spectrometer (ICPES) | Optima 7300 | Platinum Elmer USA Inc., Waltham, MA, USA | Determination of adsorption concentration |
| Dual Function Thermostat and Oscillator | SC240C | Jintan Jierier Electric Appliance Co., Jintan, China | Temperature control and shaking |
| Transmission electron microscope | Hatachi | Hitachi, Shizuoka, Japan | Characterize |
| X-ray photoelectron spectrometer | Nexsa | Thermo Fisher Scientific, Waltham, MA, USA | Determination of elemental binding states |

### 2.2. Preparation of O-MWCNT$_S$

The experiment was initiated using 100 mL of 20 wt% hydrogen peroxide mixed with 2.0000 g of MWCNT$_S$, and the solution was homogenized using magnetic stirring. Subsequently, the mixture reacted for 5 h at 333 K to produce the O-MWCNT$_S$; further, a 0.022 μm porous membrane was passed through it to obtain the powder of O-MWCNT$_S$, and the above powder was rinsed with deionized water to a of pH 7.0. Ultimately, the samples were maintained for 24 h at 353K and then allowed to cool to ambient temperature.

### 2.3. Characterization of O-MWCNT$_S$

The sample was analyzed using transmission electron microscopy (TEM) to acquire a more comprehensive understanding of the morphological changes that transpired in the MWCNT$_S$ after oxidation. Magnification in TEM ranges from 10 to 1,000,000 times, with a point resolution of 0.24 nanometers [23,24]. Furthermore, the research incorporated the utilization of an XPS to ascertain the elemental bonding states of O-MWCNT$_S$. A magnesium target with a photon energy of 1253.6 eV was employed in a monochromator for the XPS, while the laboratory vacuum was adjusted to $5 \times 10^{-9}$ Pa. And the complete XPS spectrum was utilized for qualitative analysis. Split-peak fitting was also utilized to analyze the fine spectra of C 1s (282~290 eV), O 1s (528~540 eV), Pb 4f (136~148 eV), and Cd 3d (402~416 eV) [25,26].

### 2.4. Effect of pH

In the experiment, the pH gradients for configuring ions Pb(II)/Cd(II) solutions were $2.0 \pm 0.1$, $3.0 \pm 0.1$, $4.0 \pm 0.1$, $5.0 \pm 0.1$, and $6.0 \pm 0.1$, respectively, and a fixed initial Pb(II)/Cd(II) concentration of 10 mg·L$^{-1}$ and O-MWCNT$_S$ concentration of 1.00 g·L$^{-1}$ was maintained. The temperature of the system was consistently kept at 298 K. The mixtures were put in a shaker at 200 r/min and sacrificially sampled at predetermined times; then, the samples were passed through a 25 nm membrane filter to completely remove O-MWCNT$_S$ associated with insoluble lead /cadmium species from the solution phase. The filtrates were analyzed for Pb(II)/Cd(II) remaining in the solution system with an ICP-OES.

### 2.5. Effect of Temperature

The experiments were performed at temperatures of 293, 298, and 303 K, respectively. The pH was kept at $5.0 \pm 0.1$ with HNO$_3$ or NaOH. The other experimental conditions and steps were the same as those for the effect of pH.

*2.6. Effect of Initial Concentrations*

It was only the concentrations of Pb(II)/Cd(II) metal ions that varied from 5 to 15 mg·L$^{-1}$, but the other experimental conditions and procedures were no different from the effect of pH. In addition, to study the adsorbability of O-MWCNT$_S$ for coexisting metals, the competitive test of Pb(II) and Cd(II) was conducted, as described in this subsection. It was designed such that the same mass of Pb(II) and Cd(II) was mixed in the vials and then initiated as a single system.

The following describes the structure of the experimental design utilized in this investigation:

$$\text{Removal } (\%) = (\frac{C_0 - C_e}{C_0}) \times 100 \tag{1}$$

$$\text{Adsorption capacity : } q_e = \frac{(C_0 - C_e)V}{m} \tag{2}$$

$$\text{Quasi-primary kinetic fitting : } q_t = q_e(1 - e^{-k1t}) \tag{3}$$

$$\text{Quasi-secondary kinetic equations : } q_t = \frac{K_2 q_e^2 t}{1 + K_2 q_e t} \tag{4}$$

$$\text{Langmuir isothermal equation : } \frac{C_e}{q_e} = \frac{1}{K_L q_L} + \frac{C_e}{q_L} \tag{5}$$

$$\text{Freundlich isothermal equation : } Inq_e = InK_F + \frac{1}{n}InC_e \tag{6}$$

$$\text{Arrhenius equation : } Ink_{obs} = -\frac{Ea}{RT} + InA \tag{7}$$

where $C_0$ and $C_e$ are preliminary and equilibrium concentrations, respectively (mg·L$^{-1}$); $Q_e$ denotes the adsorption amount equilibrium (mg·g$^{-1}$); Wan et al. reported that $k_1$ is a constant denoting the rate of the quasi-primary kinetic reaction, i.e., the physical adsorption rate (min$^{-1}$); and $k_2$ represents the quasi-secondary kinetic reaction rate constant, i.e., chemisorption rate (g·mg$^{-1}$·min$^{-1}$) [27]; $q_L$ is the maximum theoretical adsorption capacity (mg·g$^{-1}$); $k_L$ is Langmuir constant; $k_F$ and n are Freundlich constants; A is the Arrhenius constant; $k_{obs}$ is the rate constant [28–30].

## 3. Results and Discussion

*3.1. Characterization and Analysis of Results*

3.1.1. TEM Analysis

Figure 1 illustrates the characteristics of MWCNT$_S$ and O-MWCNT$_S$. The interwoven and agglomerated tubular arrangement of O-MWCNT$_S$ is shown in Figure 1a. The surface of the material demonstrates a relatively smooth texture, and a significant proportion of the carbon nanotube apertures appears to be closed, devoid of any discernible attachments [31]. Figure 1b shows that the material exhibits an abundance of fractures, with a higher proportion of the carbon nanotube ports being open compared to Figure 1a. Furthermore, the material exhibited a rough surface appearance as a result of the varying degrees of oxidative degradation. It is consistent with report of Wang et al. [32] that the carbon nanotubes exhibit a relatively loose structure.

3.1.2. XPS Analysis

Figure 2a presents the comprehensive X-ray photoelectron spectroscopy (XPS) spectrum of the O-MWCNT$_S$. The figure illustrates that the O-MWCNTs' surface is composed primarily of two elements, carbon (C) and oxygen (O), which accounted for 96.03% and 3.97% of the composition, respectively, and had binding energies of 286.1 and 530.6 eV. In Figure 2b, the XPS fine spectra of the C 1s peak in the O-MWCNT$_S$ are presented. The distinct binding energies of 284.8, 286.0, and 289.3 eV were attributed to the C-C, C-O-C, and O-C=O peaks, respectively. The corresponding percentages of these peaks were deter-

mined to be 55.18, 21.63, and 23.20%, respectively. Figure 2c illustrates the XPS fine spectra of the O 1s orbital in the O-MWCNT$_S$. The distinct peak patterns associated with C=O and C-O correspond to the combining energies of 531.9 and 533.4 eV, respectively. The relative percentages of these peaks have been reported as 70.47% and 29.53% [33]. Evidence for oxygen-carrying functional moieties on the surface of O-MWCNT$_S$ is provided by these observations in conjunction with Figure 2b,c.

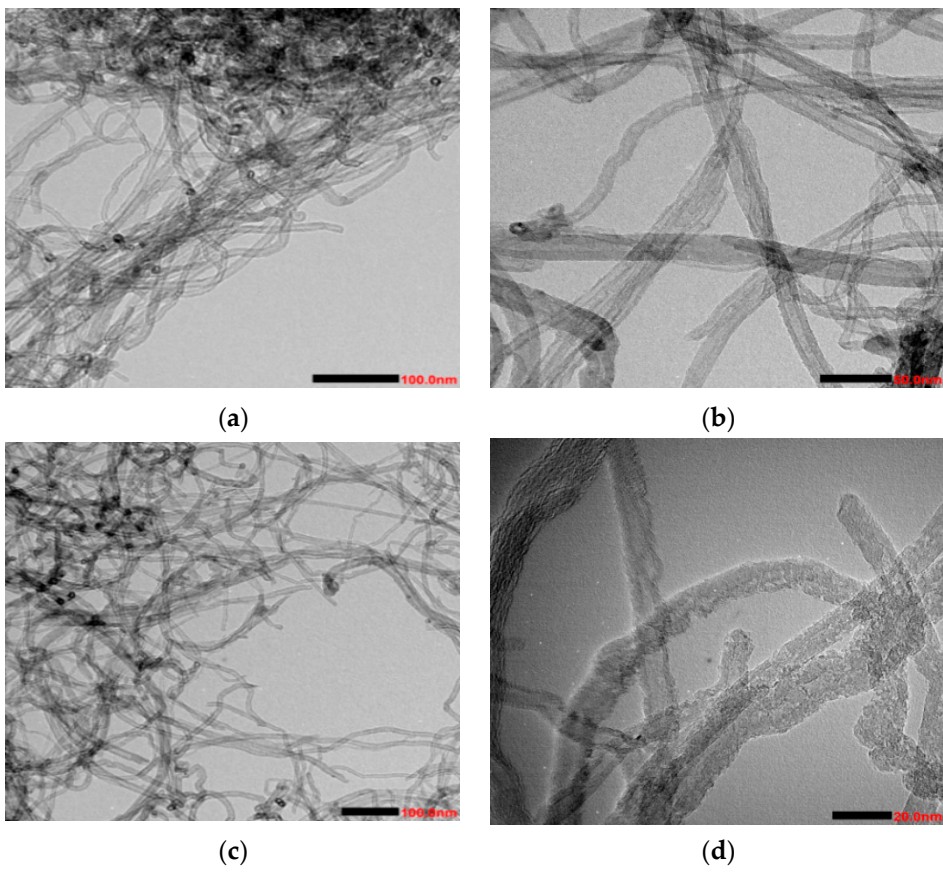

**Figure 1.** TEM analysis of MWCNT$_S$ before and after hydrogen peroxide oxidation. (**a**,**b**) Carbon nanotubes before oxidation; (**c**,**d**) carbon nanotubes after oxidation.

### 3.2. Adsorption Experiments of Pb(II)/Cd(II) with O-MWCNT$_S$

#### 3.2.1. pH

The pH affects the degree of protonation and the existence forms of Pb(II) as well as Cd(II) of O-MWCNT$_S$. Figure 3 illustrates the dynamics data of the removal of metal ions Pb(II)/Cd(II) at various pH levels (pH = 2~6) and at a stationary concentration (10 mg·L$^{-1}$) of metal ions Pb(II)/Cd(II). As illustrated in the figure, the minimum removal of Pb(II) and Cd(II) was 10.27 and 0.36% at pH 2, respectively. And the results can be explained as follows: when the pH value is less than 3.1, which is the zeta potential of O-MWCNT$_S$ [34], a substantial amount of H$^+$ is easier to occupy the adsorption sites of O-MWCNT$_S$ than metal ions Pb(II)/Cd(II). The adsorption removal of metal ions Pb(II)/Cd(II) rises to 90.15 and 31.47% with the increase in pH from 3 to 6. Importantly, at pH 6, a significantly greater amount of lead is adsorbed onto the surface of O-MWCNT$_S$. This observation indicates the strong adsorption (surface precipitation) of Pb(II) on O-MWCNT$_S$. The conclusion is in agreement with the report of Jiang et al. [35], in which it is found that as pH increased to values greater than 5.5, a fraction of Pb(OH)$_2$ appeared on O-MWCNT$_S$.

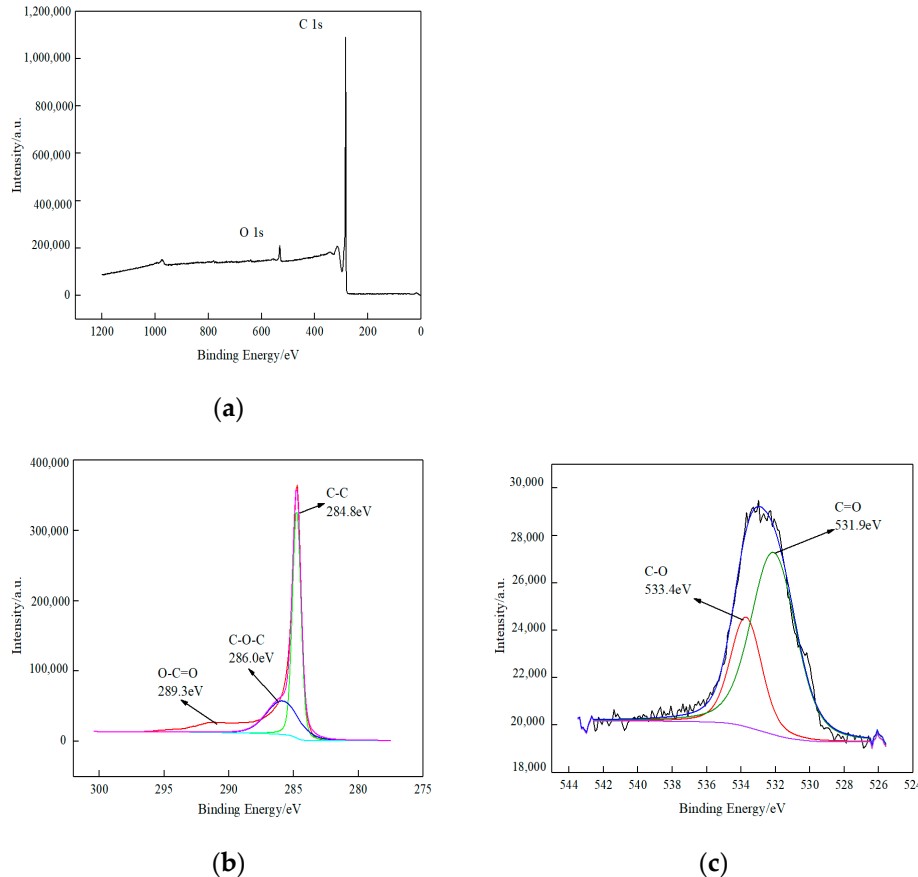

**Figure 2.** XPS spectrum of oxidized multiwalled carbon nanotubes. (**a**) XPS full spectrum of oxidized multiwalled carbon nanotubes; (**b**) fine spectrum of C 1s; (**c**) fine spectrum of O 1s.

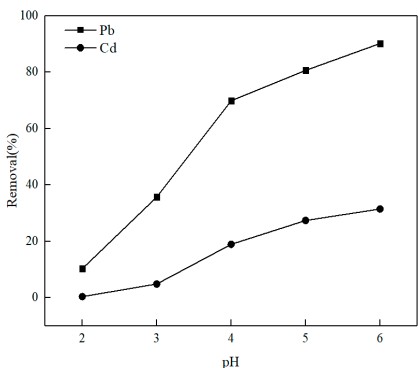

**Figure 3.** Influence of pH on adsorption removal.

### 3.2.2. Temperature

The activation energy (Ea) not only reflects the dependence of the reaction rate on temperature, but also predicts whether the reaction rate is temperature-dependent [36]. The rate constants $k_{obs}$ and the Arrhenius formula at different temperatures were utilized to determine the Ea for the adsorption of Pb(II)/Cd(II) by O-MWCNT$_S$. Fitting the logarithm of $k_{obs}$ versus $1/T$ gives a linear plot (Figure 4). The correlation coefficients of In$k_{obs}$ and $1/T$ for Pb(II)/Cd(II) were negatively correlated at 0.89 and 0.80, respectively. The activation energy Ea was calculated, using the slope, to be 83.68 kJ·mol$^{-1}$ for Pb(II) and 172.88 kJ·mol$^{-1}$ for Cd(II). From this, it follows that the adsorption of Pb(II) on the surface of O-MWCNT$_S$ is more convenient than that of Cd(II).

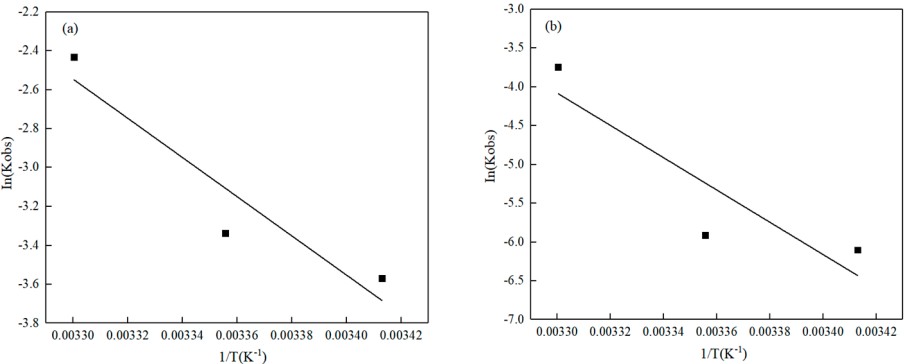

**Figure 4.** The changes in lnk with 1/T for the reaction of O-MWCNT$_S$ with Pb(II)/Cd(II). (**a**) Pb; (**b**) Cd.

### 3.2.3. Initial Concentration

Understanding the relations between the concentration of the metal ion in an aqueous solution and the quantity adsorbed by the adsorbent at equilibrium is crucial for comprehending the adsorption mechanism. The impact of the starting concentration of Pb(II)/Cd(II) on the adsorbability of O-MWCNT$_S$ was examined, as depicted in Figure 5a. The removal rate of Pb(II)/Cd(II) using O-MWCNT$_S$ decreased significantly when the starting concentrations increased. The removal rate decreased from 76.34% to 55.59% in the case of Pb(II) and from 29.83% to 16.68% in the case of Cd(II). Typically, the removal of both ions decreased with increasing initial concentration, which is attributed to the limited amount of adsorption sites for the adsorbent during the adsorption process [37]. The findings of this study suggest that Pb(II) undergoes easier adsorption onto O-MWCNT$_S$ in comparison to Cd(II). Due to the inverse relationship between the hydrated ionic radius and the hydration energy, a reduction in the hydrated ionic radius corresponds to an increase in hydration energy. Consequently, the removal efficiency of metal ions increases with a decrease in hydrated ionic radius. The results of this experiment demonstrated that the hydration ion radius of Pb(II) is comparatively greater, resulting in a significantly higher removal rate. The role of hydration energy is higher than that of hydration ion radius.

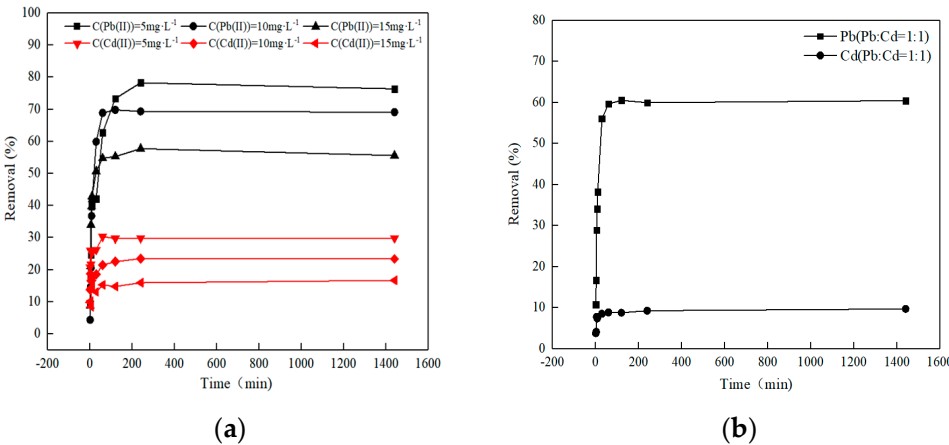

**Figure 5.** (**a**) Effect of initial concentration on Cd(II) and Pb(II) adsorption in a single system. (**b**) Adsorption experiments with Pb(II):Cd(II) = 1:1.

The results of the competitive tests are shown in Figure 5b. After adsorption equilibration was reached, the removal of Pb(II) was about 60.46%, while that of the other metal ion, Cd(II), was 9.70%. When the removal rate is compared to individual adsorption, a slight decrease in quantity is observed. This observation may indicate an antagonistic effect in the sorption of Pb(II) and Cd(II) on the surface of O-MWCNT$_S$.

The results of fitting the Langmuir and the Freundlich isothermal curves to the experimental data at 298 K are displayed in Figure 6, and the parameters of the models are indicated in Table 2. Table 2 demonstrates that the Langmuir isothermal models could better describe the adsorption of both ions, according to an $R^2$ of 0.99 with Pb(II) and 0.91 with Cd(II). This outcome suggests that chemical adsorption was the major reaction [38,39]. The linear Langmuir model yielded the highest absorption capabilities: 5.73 mg·g$^{-1}$ with Pb(II) and 3.34 mg·g$^{-1}$ with Cd(II).

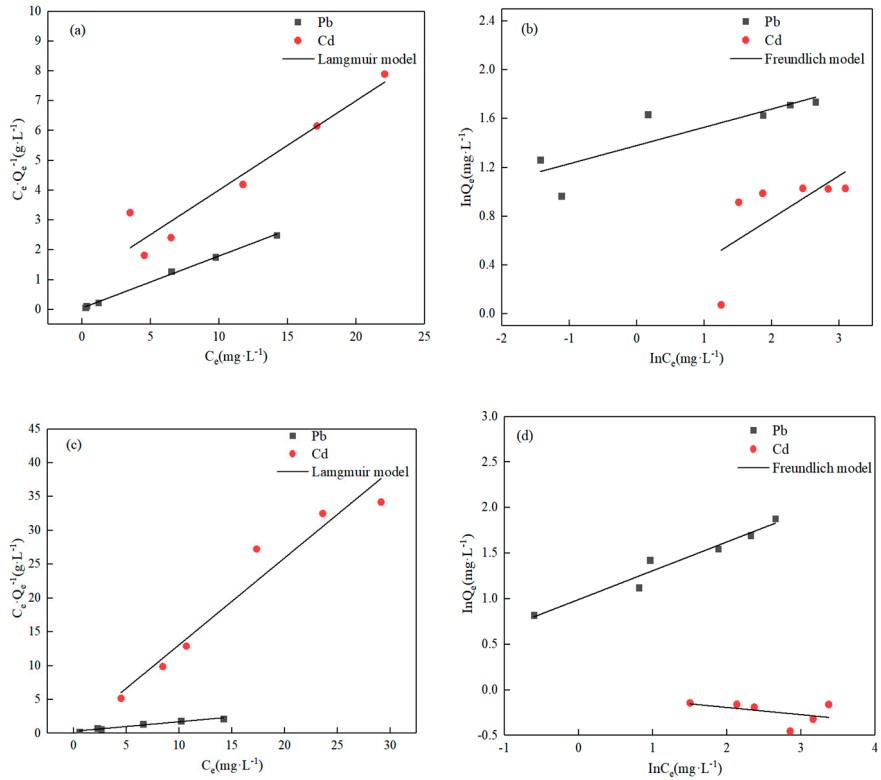

**Figure 6.** Adsorption isothermal linear fitting. (**a**,**b**) Single system; (**c**,**d**) competitiveness system.

**Table 2.** Parameters fitted to the adsorption process using the Langmuir and Freundlich models.

| Adsorption Conditions | Heavy Metal Ions | Langmuir | | | Freundlich | | |
|---|---|---|---|---|---|---|---|
| | | $Q_m$ (mg·g$^{-1}$) | $K_L$ (L·mg$^{-1}$) | $R^2$ | $n_F$ | $K_F$ | $R^2$ |
| Single | Pb(II) | 5.73 | 3.12 | 0.99 | 6.67 | 3.97 | 0.73 |
| | Cd(II) | 3.34 | 0.29 | 0.91 | 2.86 | 1.09 | 0.47 |
| Competitiveness | Pb(II) | 7.11 | 0.44 | 0.96 | 3.13 | 2.69 | 0.95 |
| | Cd(II) | 0.78 | 4.84 | 0.94 | −12.5 | 0.97 | 0.2 |

Note: Here is only listed the parameters of the reaction temperature of Pb/Cd as 298 K and the competitive adsorption of Pb and Cd.

The sorption isotherm results for the competing systems are consistent with the single system. The linear Langmuir model fitted the highest uptake capacity of 7.11 mg·g$^{-1}$ in the case of Pb(II), and 0.78 mg·g$^{-1}$ in the case of Cd(II). It can be concluded that it is easier to absorb Pb(II) than Cd(II) on the O-MWCNTs' surface, and this result is in mutual agreement with the activation energy calculations.

Table 3 lists the fitting parameters associated with quasi-primary and quasi-secondary kinetics for the experimental study. All the $R^2$ values of the quasi-primary kinetics and quasi-secondary kinetics are greater than 0.80. That is to say, both models can better describe the data. A more precise depiction of the adsorption process of metal ions Pb(II)/Cd(II) is offered by both the quasi-primary and quasi-secondary kinetic models [40–43]. Upon

comparing $k_1$ and $k_2$, it is evident that Pb(II) demonstrates a greater rate of physisorption than Cd(II) in both single and competitive systems ($k_{Pb1} > k_{Cd1}$), but, on the contrary, the chemisorption rate of Cd(II) is greater than that of Pb(II) ($k_{Cd2} > k_{Pb2}$) [27,44,45].

**Table 3.** Adsorption kinetics fitting parameters.

| Adsorption Conditions | Metal Type | Quasi-Primary Kinetic Fitting | | | Quasi-Secondary Kinetic Fitting | | |
|---|---|---|---|---|---|---|---|
| | | $k_1$ | $Q_{e1}$ | $R^2$ | $k_2$ | $Q_{e2}$ | $R^2$ |
| Single | Pb | 0.1 | 6.89 | 0.99 | 0.01 | 7.41 | 0.97 |
| | Cd | 0.08 | 2.46 | 0.95 | 0.05 | 2.65 | 0.96 |
| Competitiveness | Pb | 0.11 | 5.97 | 0.98 | 0.02 | 6.33 | 0.98 |
| | Cd | 0.01 | 1.07 | 0.83 | 0.45 | 1.12 | 0.88 |

Note: Here is only listed the parameters of the concentration of Pb/Cd as 10 mg·L$^{-1}$ and competitive adsorption.

### 3.3. Adsorption Mechanism Analysis

The mechanism of O-MWCNTs' adsorption of Pb(II)/Cd(II) was determined using XPS [46–48]. Figure 7 shows the XPS spectra of the powdered samples of the fresh and spent O-MWCNT$_S$. Figure 7a displays the fine spectra of the C 1s orbital for the spent O-MWCNT$_S$. The peak of C-C, C-O-C, and O-C=O has a combining energy of 284.8, 286.0, and 289.3 eV, respectively, which clearly indicates that an oxygen-containing functional group is in the C-O-C and O-C=O states expected for the native oxide of MWCNT$_S$. The figures presented in Figure 7b,c illustrate the distinctive peaks of C 1s of the spent O-MWCNT$_S$. The combining energies of C-O-C and O-C=O are observed to be lower than those illustrated in Figure 7a. The observed decrease in binding energies implies that the oxygen-containing functional groups (C-O-C and O-C=O) reacted with Pb(II)/Cd(II). Figure 7d,e shows O-Pb and O-Cd spectra collected from the surface of the power sample. The combined energy of O-Pb and O-Cd is 534.5 eV and 535.1 eV, respectively. Figure 7f provides further evidence that the adsorption process is facilitated with the C-O and C=O groups present. Figure 7g shows that the Pb 4f binding energy is 138.9 eV and 143.8 eV. Similarly, the binding energy of Cd 3d is observed at 406.1 and 412.7 eV in Figure 7h. These findings suggest the successful adsorption of Pb(II)/Cd(II) onto O-MWCNT$_S$. In contrast to the conventional Pb 4f profile, the binding energy shifts to slightly higher sites. In addition, the binding energies of these peaks are in agreement with those reported by Zhan et al. and Abdelmoula et al. [49].

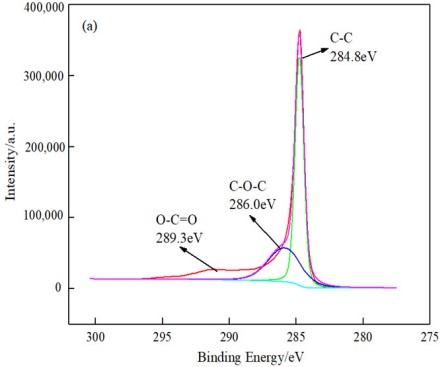 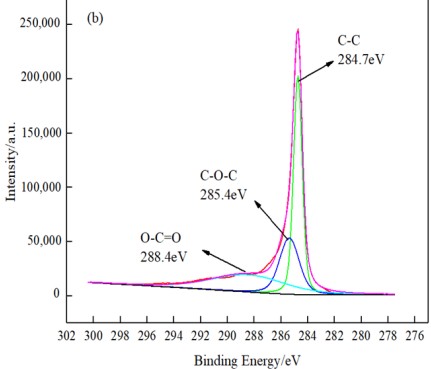

**Figure 7.** *Cont.*

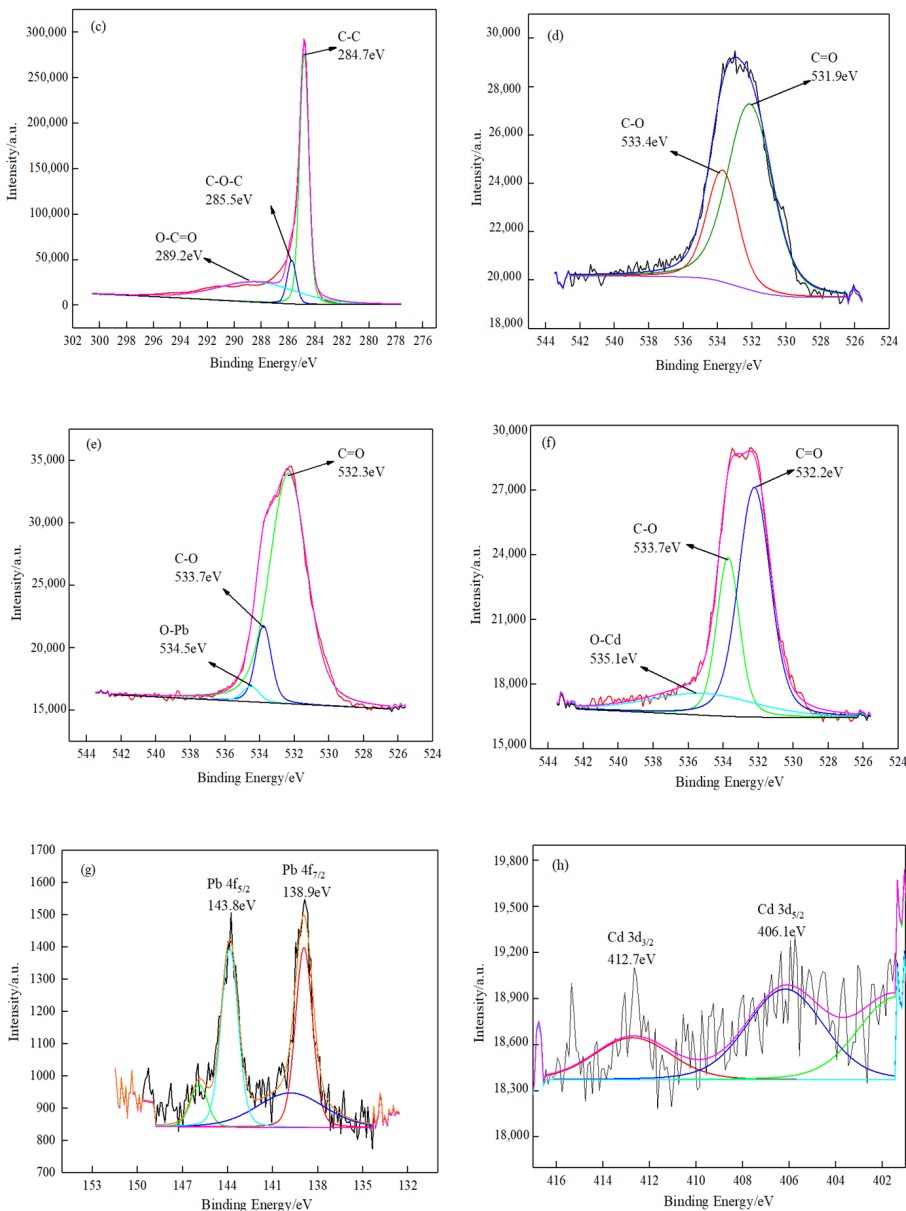

**Figure 7.** XPS fine spectra of O-MWCNT$_S$ before and after the adsorption of metal ions Pb(II) and Cd(II). (**a–c**) The C 1s spectra of O-MWCNT$_S$; O-MWCNTs' adsorption of Pb(II) and Cd(II); (**d–f**) the O 1s spectra of the three scenarios; (**g**) the Pb 4f spectrum of O-MWCNTs' adsorption of Pb(II); (**h**) the Cd 3d spectrum of O-MWCNTs' adsorption of Cd(II).

Based on the research conducted on the basis of XPS, the adsorption mechanism of O-MWCNT$_S$ may be inferred [50].

$$MWCNT_S + H_2O_2 \rightarrow MWCNT_S\text{-}COO^- + MWCNT_S\text{-}O^-$$
$$2(\text{-}COO^-) + Pb^{2+} \leftrightarrow (\text{-}COO)_2Pb$$
$$(\text{-}COO^-) + (\text{-}O^-) + Pb^{2+} \leftrightarrow (\text{-}COO)Pb(\text{-}O)$$
$$2(\text{-}COO^-) + Cd^{2+} \leftrightarrow (\text{-}COO)_2Cd$$
$$(\text{-}COO^-) + (\text{-}O^-) + Cd^{2+} \leftrightarrow (\text{-}COO)Cd(\text{-}O)$$

## 4. Conclusions

The results of the experiment demonstrated the potential of O-MWCNT$_S$ for the absorptive removal of Pb(II)/Cd(II) from water environments. The major conclusions from this study are summarized as follows:

- Batch kinetic experiments indicated that O-MWCNT$_S$ can effectively absorb metal ions Pb(II)/Cd(II) in water. The batch kinetic tests showed that a pH of 6 was most favorable for the absorption of Pb(II)/Cd(II). When the pH was increased to 6, the removal rate of Pb(II) and Cd(II) was increased to 90.15 and 31.47%, respectively. The removal rate of 76.34% decreased to 55.59% for Pb(II) and from 29.83% to 16.68% for Cd(II), when the starting concentration of Pb(II)/Cd(II) ranged from 5 to 15 mg·g$^{-1}$. The removal rate in the competitive tests was about 60.46% with Pb(II) and 9.70% with Cd(II), and the tests proved that Pb(II) is more easily adsorbed on the surface of the O-MWCNT$_S$. The Langmuir model was better at describing the absorptive data for both ions. And the Q$_m$ of Pb(II) was 5.73 mg·g$^{-1}$, while that of Cd(II) was 3.34 mg·g$^{-1}$ in the single-ion system; the Q$_m$ was 7.11 mg·g$^{-1}$ with Pb(II) and 0.78 mg·g$^{-1}$ with Cd(II) in the competitive system.
- Based on XPS analyses, it can be summarized that the absorbed lead/cadmium species on the surface of the O-MWCNT$_S$ was (-COO)$_2$Pb and (-COO)Pb(-O)/(-COO)$_2$Cd and (-COO)Cd(-O).
- Thermodynamic tests indicated that the activating energy was 83.68 kJ·mol$^{-1}$ for Pb(II) and 172.88 kJ·mol$^{-1}$ for Cd(II). The adsorption of Pb(II) on the surface of O-MWCNT$_S$ was more convenient than that of Cd(II).

**Author Contributions:** Conceptualization, X.L. and H.L.; methodology, X.L. and H.L.; validation, X.L., Y.C. and W.D.; formal analysis, X.L. and W.C.; investigation, H.L. and L.H.; resources, H.L.; writing—original draft preparation, X.L. and H.L.; writing—review and editing, X.L. and H.L. All authors have read and agreed to the published version of the manuscript.

**Funding:** This research was funded by the National Natural Science Foundation of China: 41807130.

**Institutional Review Board Statement:** Not applicable.

**Informed Consent Statement:** Not applicable.

**Data Availability Statement:** The data presented in this study are available on request from the corresponding author.

**Conflicts of Interest:** The authors declare no conflicts of interest.

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
