# Peer review of "Adsorption Kinetics and Mechanism of Pb(II) and Cd(II) Adsorption in Water through Oxidized Multiwalled Carbon Nanotubes"

_applsci, doi:10.3390/app14051745_

Round 1
Reviewer 1 Report
Comments and Suggestions for Authors
See the attached file.

Need to be revised.
Reviewer 2 Report
Comments and Suggestions for Authors
This paper is an interesting approach to removing heavy metals from water using inorganic materials. The reviewer finds the paper suitable to be accepted after some minor improvements (the reviewer is willing to review the revised version of this paper), as follows:
1) Discuss the role of inorganic materials (like oxidized carbon nanotubes) in the adsorption of heavy metals when incorporated in membranes. Somehow, authors can also discuss in terms of superior properties (thermal and physicochemical in membranes). Please, read, discuss and cite accordingly these updated highly cited papers:
-Ongoing progress on novel nanocomposite membranes for the separation of heavy metals from contaminated water
-A critical review on electrospun membranes containing 2D materials for seawater desalination
2) compare your results with other nanomaterials reported in the literature.
3) what is next in this research? future perspective of this material? maybe membranes? please, guve us a feedback
Reviewer 3 Report
Comments and Suggestions for Authors
In the manuscript with the title “Adsorption kinetics and mechanism of Pb(II) and Cd(II) adsorption in water by oxidized carbon nanotubes“, the authors studied the adsorption characteristics of O-MWCNTS on Cd(II) as well as Pb(II), and the competitive adsorption relationship between these metals.
The paper is clearly presented and provides interesting results. However, it can be accepted for publication after incorporating the following comments.
1. Introduction
Bearing in mind, that a large number of papers have been published on the topic of heavy metal adsorption the introduction is written too generally and without a serious review of the literature.
2. Materials and Methods
In Table 1 is necessary to insert the column for which type of analysis the specified instrument was used.
3. Results and Discussion
In section 3.2 Adsorption experiments of Pb(II) and Cd(II) by O-MWCNTS
Max adoption capacity needs to be added for both metals
Line 206-207 “On the contrary, the chemisorption rate of Cd(II) is greater than that of Pb(II) (kCd2 > kPb2)“. , insert reference.
Figure 5. In order to clearly see the results, it would be good if the scale on the y-axis were the same for all diagrams
In section 3.3 Adsorption mechanism analysis,- no explanation for physisorption which was confirmed to exist in the previous section.
It would also be nice to explain the mechanism through isotherms and see if there is a difference in the adsorption results.
In order to clearly see their contribution, authors must better explain why their work is significant. It is suggested that based on comparing the research experiments of other researchers, the author objectively evaluates and analyzes whether this research method is reliable and what the basis, it is should be written clearly.
In addition, for example, if natural zeolite adsorbs 30-60%mg/g Pb and O-MWCNTS 6mg/g, why would this be done and how is it economically profitable?
4. Conclusion
The conclusion should be rewritten following the changes requested above.
Round 2
Reviewer 3 Report
Comments and Suggestions for Authors
After reading the revised review paper previously titled ‘’Adsorption kinetics and mechanism of Pb(II) and Cd(II) adsorption in water by oxidized multiwalled carbon nanotubes” I can say that the authors successfully responded to all the requests addressed to them and significantly improved the work, which in this form confirms the validity of their results.
For this reason, I recommend this paper for publication in its present form.
Author Response
Dear Reviewer,
Thank you for reviewing our manuscript and providing your valuable comments and suggestions. We appreciate you taking your valuable time to review our manuscript. Your suggestions have enabled us to improve our work.
Thank you again for your time and professional input!
Yours sincerely,
Xin Li.